# Sexuality as a Prognostic Factor—Results of an Individual Patient Data NOGGO (North-Eastern German Society of Gynecological Oncology)-Meta-Analysis of 644 Recurrent Ovarian Cancer Patients Prior to Chemotherapy

**DOI:** 10.3390/cancers16040811

**Published:** 2024-02-16

**Authors:** Nicole Balint, Hannah Woopen, Rolf Richter, Adak Pirmorady-Sehouli, Klaus Pietzner, Jalid Sehouli

**Affiliations:** 1Department of Gynecology with Center for Oncological Surgery, Campus Virchow Klinikum, Charité-Universitätsmedizin Berlin, Corporate Member of Freie Universität Berlin and Humboldt Universität zu Berlin, Augustenburger Platz 1, 13353 Berlin, Germanyelro.richter@online.de (R.R.); klaus.pietzner@charite.de (K.P.); jalid.sehouli@charite.de (J.S.); 2Department of Psychosomatic Medicine, Charité-Universitätsmedizin Berlin, Corporate Member of Freie Universität Berlin and Humboldt Universität zu Berlin, Hindenburgdamm 30, 12203 Berlin, Germany; adak.pirmorady@charite.de

**Keywords:** ovarian cancer, interest in sex, sexual activity, quality of life, discontinuation of chemotherapy, survival

## Abstract

**Simple Summary:**

The topic of sexuality in patients with ovarian cancer is still insufficiently addressed in both science and clinical practice. The aim of our retrospective study was to investigate the association between the sexuality of patients with recurrent ovarian cancer before chemotherapy and patient characteristics such as quality of life, treatment discontinuation, and survival. We demonstrate that patients who were interested in sex and were sexually active had the best prognosis. Thus, health care professionals should routinely address sexuality in patients with ovarian cancer because sexuality appears to be a marker of quality of life and overall survival. This should be validated by further prospective analyses.

**Abstract:**

Background: The aim of this study was to analyze the associations between sexuality, quality of life, treatment discontinuation, and survival in recurrent ovarian cancer (OC). Methods: Raw data from various phase II/III studies, including the questionnaires EORTC-QLQ-C30 and QLQ-OV28, were included. Data from the meta-analysis were calculated using logistic and Cox regression. Results: Data on sexuality were available for 644 patients. A total of 162 patients had an interest in sex and were sexually active (Group A). A total of 45 patients had an interest in sex and were sexually not active (Group I) and 437 patients had no interest in sex and were not sexually active (Group N). Group A was younger in median age (age at randomization), at 57 years, than Group I, at 60 years, and Group N, at 65 years (*p* < 0.001). Group A had a better ECOG performance status and fewer recurrences (all *p* < 0.001). FIGO stage, grading, and BMI were not associated with interest in sex and sexual activity. Group A showed higher scores in role, body, and social function (all *p* < 0.001), emotional functionality (*p* < 0.002), and body image (*p* = 0.012). In addition, Group A reported less pain, less peripheral neuropathy, and less fatigue (all *p* < 0.001). There was no association with the premature discontinuation of chemotherapy. Group A showed better survival rates compared to group N (22.3 months vs. 17.4 months, *p* < 0.001). Conclusions: Physicians should routinely address the topic of sexuality with ovarian cancer patients. Sexuality appears to be a marker for quality of life as well as overall survival.

## 1. Introduction

Treatment regimens for patients with ovarian cancer are intensive and include extensive multivisceral surgery, multiple chemotherapies, and maintenance therapies [1]. However, there is little information on the impact of surgical and adjuvant therapies on the sexuality of ovarian cancer patients [2]. We know from the Lion study, which investigated the impact of lymphadenectomy in ovarian cancer patients, that the surgical procedure can negatively impact sexuality and patients seem to be less interested in sexual activities due to multimodal cancer therapy (surgery and chemotherapy) [3].

The sexual dysfunction reported by ovarian cancer patients most commonly includes pain during intercourse (dyspareunia), decreased sexual desire, and problems with arousal or achieving orgasm [2,4,5].

In an evaluation of 20 long-term ovarian cancer survivors with an average age of 68 years, women were found to have low rates of sexual desire. Problems with vaginal lubrication, orgasm, and dyspareunia were reported by the affected patients. All long-term survivors had received surgical cytoreduction, while 75% of them had experienced a recurrence. [6].

With the advances in systemic therapy and surgical techniques in recent decades, more than 43% of affected women across all stages of ovarian cancer survive the first five years [7]. Therefore, the quality of life of affected individuals is of particular importance [8]. There are inconsistent results regarding the quality of life and sexuality of women with ovarian cancer [2,4,9,10,11,12,13,14].

In order to comprehend the complexity of sexuality, women must be perceived and seen as a whole biopsychosocial entity [15,16]; thus, sexuality can be understood as a biologically, psychologically, and socially [17] determined quality of human experience [15]. Research shows that survivors of gynecological cancers have several psychological concerns regarding their femininity [18] attributed to changes in body image, such as scars, weight changes, hair loss, loss of their reproductive organs, and lymphedema [14,19]. In the research group of Nowosielski et al., a pilot project that took place after cancer diagnosis in women showed that the sexuality of her sexual partner can also be affected [19]. We also know from the research that women with ovarian cancer may experience reduced partner intimacy [20].

Social stress was reported due to women’s changing roles in the family and in social relationships [21]. Women with ovarian cancer are usually not adequately informed about the effects of the cancer treatment on their sexuality [14]. Communication between the treating physician, the patient, and her partner is mostly about the upcoming surgical or drug therapies and tumor follow-up, rarely about the impact of the diagnosis and treatments on the patient and partner’s sexuality [20]. Possible reasons for this may be an inadequate education of the medical personnel [13], who commonly wait until the affected person raises the issue herself [14]. In many studies, medical staff are sensitive to discussions of changes in sexuality with women affected by ovarian cancer [2,4,5,13,14,19,22], such as decreased sexual desire, vaginal dryness, and pain during penovaginal intercourse. We agree that open and supportive communication should be offered to patients and their partners about ovarian cancer’s impact on their sexuality [20]. The topic of sexuality in patients with ovarian cancer is insufficiently addressed, both in clinical practice and in the scientific community [23].

It remains unclear whether decreased sexual activity is causative or correlates with other independent prognostic factors, such as tumor stage or therapy-specific symptoms. In our analysis, we would like to investigate the prognostic significance of sexual activity on patient survival, which has remained scientifically unstudied.

## 2. Materials and Methods

For this individual-participant-data meta-analysis, the following four different phase II/III clinical trials, conducted by the North-Eastern German Society of Gynecological Oncology (NOGGO), were integrated in one large meta-database. All four of the aforementioned phase II/III studies were reviewed by the local ethics committees (Hector: LAGeSo AVD-No. ZS 14 627/06; Tower: Charité-Universitätsmedizin Berlin AVD-No. EA 2/115/05; TRIAS: LAGeSo AVD-No. ZS EK 10; Treosulfan: Charité-Universitätsmedizin Berlin AVD-No. 145/2004).

### 2.1. Tower Study

This study was a phase 2 trial with a topotecan weekly schedule versus the conventional 5-day treatment schedule [24]. Patients with recurrent platinum-resistant ovarian cancer after radical surgery and platinum chemotherapy were selected for this study between September 2005 and February 2008.

### 2.2. TRIAS Study

The TRIAS study was conducted between January 2010 and September 2013 [25]. This particular study included 185 elderly patients with recurrent ovarian cancer. Furthermore, the study was a randomized, double-blind, placebo-controlled, phase 2 trial where patients were assigned a topotecan with Sorafenib or a placebo.

### 2.3. Hector Study

Between March 2007 and December 2009, 550 patients with platinum-sensitive ovarian, peritoneal, and fallopian tube carcinoma were selected for this phase 3 study [26]. Patients were randomly assigned to receive either a standard platinum-based combination or Topotecan and Carboplatin therapy.

### 2.4. Treosulfan Study

This particular study was carried out between August 2004 and December 2010 and included 123 elderly patients with recurrent ovarian cancer [27]. Patients could decide whether they wanted to take the therapy orally at home or have it administered intravenously.

### 2.5. EORTC questionnaires QLQ-C30 and EORTC QLQ-OV28

In these trials, quality of life was assessed with the EORTC questionnaires QLQ-C30 [28] (Version 3; “Hector” and “TRIAS”) and EORTC QLQ-OV28 [29] (“Hector”, “Tower”, “Trias” and “Treosulfan”). The European Organization for Research and Treatment of Cancer (EORTC) QLQ-C30 is a 30-item questionnaire. It includes a global health and quality of life scale, three symptom scales, five functional scales, and six single items. The three symptom scales comprise nausea/vomiting, fatigue, and pain. Among the five functioning scales, emotional, social, cognitive, physical, and role functioning are measured. Several individual symptom measures are also included in the EORTC QLQ-C30. Individual symptom measures include loss of appetite, diarrhea, constipation, dyspnea, insomnia, and financial difficulties. The EORTC QLQ-C30 questionnaire is a widely used quality of life instrument for several cancer patient groups, including ovarian cancer patients, that has been validated in terms of its reliability and sensitivity, and provides consistent results [30].

The EORTC QLQ-OV28 ovarian cancer module is a validated questionnaire [20]. It additionally contains 28 questions assessing specific disease- and treatment-related symptoms to measure the aspects of quality of life of patients with ovarian cancer that are not covered by the EORTC QLQ-C30 questionnaire. The questionnaire (EORTC QLQ-OV28) was developed as a supplement to the EORTC QLQ-C30 questionnaire. The Ovar28 questionnaire includes questions on abdominal/gastrointestinal symptoms, peripheral neuropathy, other chemotherapy-related side-effects, hormonal/menopausal symptoms, body image, and attitude to disease and treatment, as well as four questions on sexuality during the past four weeks. The definition of interest in sex and sexual activity was based on questions 55 (“To what extent were you interested in sex?”) and 56 (“To what extent were you sexually active?”). Questions 57 (“To what extent was sex enjoyable for you?”) and 58 (“Did you have a dry vagina during sexual activity?”) were only to be filled out if patients were sexually active. Based on interest in sex and sexual activity, three groups were defined. Group A (A = active), Group I (I = interest) and Group N (N = not).

### 2.6. Statistics

The statistical program IBM SPSS Statistics 25 (SPSS Inc., an IBM Company, Chicago, IL, USA) was used for all statistical analyses. *p* < 0.05 was considered significant. Chi-square and Kendall’s tau-b were used to compare clinical characteristics between the three sexuality groups. Differences between the three sexuality groups regarding quality of life and symptoms were detected with ANOVA. For prior discontinuation of therapy, odds ratios were calculated with logistic regression analyses after adjusting for covariates. Kaplan–Meier was used to estimate median survival. Hazard ratios for overall survival were calculated with Cox regression after adjusting for covariables.

## 3. Results

Altogether, 644 recurrent ovarian cancer patients were included in this analysis. The majority of patients (88.6%) were diagnosed with advanced-stage ovarian cancer (FIGO III/IV) at initial diagnosis. At randomization, 64.6% had their first recurrence, 25.5% their second recurrence and 9.9% of patients had more than two recurrences. Almost half of the patients (46.5%) showed a very good performance status (ECOG 0), 46.8% had an ECOG 1, and 6.7% had ECOG 2. The definition of interest in sex and sexual activity was based on questions 55 and 56 in the EORTC questionnaires’ Ovar28. The Ovar28 was available to 802 patients. Both question 55 (“To what extent were you interested in sex?”) and question 56 (“To what extent were you sexually active?”) were answered by 644 patients. Our patient cohort was divided into three groups: (A = active) patients who had an interest in sex and who were sexually active (n = 162), (I = interest) patients who had an interest in sex but were not sexually active (n = 45), and (N = not) patients who had no interest in sex and were not sexually active (n = 437).

### 3.1. Association with Patients’ Characteristics

Patients who were interested in sex and who were sexually active were significantly younger, with a median age of 53 years, compared to patients who were not active (group I: 58 years, group N: 62 years), *p* < 0.001. Interest in sex and sexual activity were also significantly associated with ECOG performance status (*p* < 0.001), number of recurrences (*p* < 0.001), and ascites (*p* = 0.002). Grading, FIGO stage, and BMI were not associated with sexual interest and activity (Table 1).

### 3.2. Association with Quality of Life

Sexual interest and activity were correlated with global quality of life, functionalities, and symptoms based on the EORTC quality of life questionnaire QLQ-C30 (Figure 1) and EORTC QLQ-OV28 (Figure 2). Data for the EORTC QLQ-OV28 were available from 644 patients. Global quality of life was higher in patients who had an interest in sex and who were sexually active and was lowest in patients who were not sexually interested and not sexually active (*p* < 0.001). Regarding the different functionalities of quality of life, patients in group A (sexually interested and active) showed higher levels in terms of physical functionality (*p* < 0.001), social functionality (*p* < 0.001), role functionality (*p* < 0.001), and emotional functionality (*p* = 0.002). Group A reported less nausea and vomiting (*p* = 0.007), less loss of appetite (*p* < 0.001), less diarrhea (*p* = 0.008), less dyspnea (*p* < 0.001), less pain (*p* < 0.001), less fatigue (*p* < 0.001), and less insomnia (*p* = 0.004). There were no differences regarding constipation and financial difficulties between the three groups. The analyses of the Ovar28 showed that patients who were interested in sex and sexually active (group A) reported lower levels of gastrointestinal symptoms (*p* < 0.001) and peripheral neuropathy (*p* < 0.001). Hormonal symptoms like hot flushes or night sweats did not differ between patient groups. Patient group A showed the lowest levels for attitude to disease and treatment compared to the other groups (*p* = 0.046). Patients in group I and N felt less attractive and were more dissatisfied with their body (body image: *p* = 0.012) (Figure 1 and Figure 2).

### 3.3. Discontinuation of Chemotherapy

In the entire patient cohort, 27.9% discontinued chemotherapy earlier than planned. In univariate analysis, there was an association of sexual interest and activity with discontinuation of treatment (*p* = 0.031). Nonetheless, in multivariate analyses, this finding could not be confirmed. Other factors, such as age at randomization, number of recurrences, and the presence of ascites, were more important.

### 3.4. Association with Survival

Follow-up data, including survival data, were available for 642 patients. Median overall survival was 19.4 months for the entire patient group of 642 patients. Patients who were interested in sex and were sexually active showed better survival rates compared to patients who were not interested and not sexually active (22.3 months vs. 17.4 months, *p* < 0.001 in pairwise comparisons). The median survival of patients who were interested in sex but not sexually active was 27.4 months. Overall survival was associated with interest in sex and sexual activity. Overall survival was significantly shorter in patients without interest in sex who were not sexually active (hazard ratio (HR) 1.5; 95% confidence interval (CI) 1.19–1.92; *p* = 0.001). This finding could be confirmed via multivariate cox regression analysis after adjusting for age at randomization, number of relapses, FIGO, and ECOG (HR 1.36, 95% CI 1.05–1.76, *p* = 0.02) (Figure 3).

## 4. Discussion

In this individual data–NOGGO meta-analysis, 644 patients with recurrent ovarian cancer were studied before starting chemotherapy. It was found that patients who were interested in sex and sexually active were younger, and had a higher quality of life and higher functioning. There was no association with premature discontinuation of chemotherapy. Overall survival was associated with interest in sex and sexual activity.

This is the first time that sexuality has been identified as a prognostic marker in women with recurrent ovarian cancer. Despite the retrospective design of the analysis, the fulfillment of the inclusion and exclusion criteria, and the related selection bias, we consider this study clinically and scientifically relevant to this taboo topic.

A possible explanation for the results is tumor- and therapy-related symptoms such as pain, polyneuropathy, fatigue, gastrointestinal symptoms, nausea, and vomiting. Our analysis showed that women who were interested in sex and were active had less pain. For example, in our group’s analysis by Woopen et al. [31], pain was shown to have a negative impact on quality of life and was associated with decreased survival, and the treatment of pain with pain management influenced overall survival. Our study indicated that women who were interested in sex and were active reported less fatigue. In the pooled meta-analysis with over 3900 patients by Sloan et al. [32], fatigue was shown to be an independent and strong prognostic factor for overall sur-vival in different oncology patient populations which went beyond physical statutes. One possible treatment approach is physical activity interventions [33,34,35]. Our analysis presented that women who were interested in sex and were active reported fewer neuropathy symptoms. A prospective study by Bennedsgaard et al. [36] analyzed patients with breast cancer and colorectal carcinoma, suggesting that treatment with docetaxel and oxaliplatin could lead to neuropathic pain and polyneuropathy, which were common long-term side effects that had a negative impact on quality of life.

A possible treatment approach to improve the functional limitations of polyneuropathy could be achieved by non-drug therapies such as sports therapy, physiotherapy, and occupational therapy [37,38]. However, data regarding the connection between polyneuropathy and sexuality are limited. Our research showed that women who were interested in sex and were active reported fewer gastrointestinal symptoms. Also, these women reported less diarrhea and less loss of appetite. The Aurelia sub-study showed that women with platinum-resistant ovarian cancer who had low abdominal/intestinal symptom scores showed longer survival than affected women who had a higher symptom burden such as ascitis and peritoneal carcinomatosis. Thus, they concluded that physical functioning and gastrointestinal/abdominal symptoms may predict overall survival in women with platinum-resistant ovarian cancer [30]. Nevertheless, specific data on sexuality were not represented. In our analysis, the women who were interested in sex and were active had less nausea and vomiting. In the study of Quinten et al. [39], vomiting and nausea were found to be associated with survival in women with ovarian cancer in the EORTC QLQ-C30. We can speculate that the best supportive care might improve women’s sexuality. However, this study underlines the high relevance of providing the best supportive care therapy regarding antiemesis and pain therapy.

In addition, sexuality could also be a surrogate marker for general well-being. This hypothesis should be evaluated in future projects.

No psycho-oncological intervention was performed in our study, but we would recommend that one is carried out in prospective studies to examine its influence. In our analysis, higher body image scores were obtained when women had an interest in sexuality and were sexually active. Women who had no interest and were not active felt less physically attractive as a result of their disease or treatment, and were more dissatisfied with their bodies. Providing psycho-oncological support to strengthen body image, emotional and social functionality could possibly bring about a change in sexual empowerment.

Women who had an interest in sexuality and were active were significantly younger and in better general health. This could have had a substantial impact on their overall survival. Future studies should evaluate whether the effect we described is also seen in specific subgroups. This concerns, for example low- and high-grade platinum re-induction and platinum-free therapy, as well as targeted maintenance therapy. We did not have a central pathology that could subsequently adapt the change to the new WHO classification of 2014 [40] and the FIGO classification of 2014 [40,41]. Furthermore, the two-part classification of platinum-resistant and platinum-sensitive was changed in 2017 [42,43]. We were unable to make these distinctions in our database.

Because we did not inquire as to relationship status are other sociocultural aspects, we recommend that this is determined in future prospective studies to possibly learn more about the patient group who were interested in sex but not sexually active. Also, our questions about sexuality referred to the last four weeks of a patient’s life. Longitudinal studies are necessary to map sexuality before and after therapy, including cancer care follow-up.

## 5. Conclusions

Sexuality seems to be a marker for quality of life, as well as for overall survival. Moreover, our study demonstrates that patients who were interested in sex and were sexually active were also better equipped with their own management of the disease. These patients reported fewer gastroenterological problems (such as nausea, vomiting, diarrhea, and loss of appetite), and higher levels of social and psychological functions. Therefore, we recommend that health care professionals routinely address sexuality in patients with ovarian cancer. Future studies are required to confirm the prognostic role of sexuality.

## Figures and Tables

**Figure 1 cancers-16-00811-f001:**
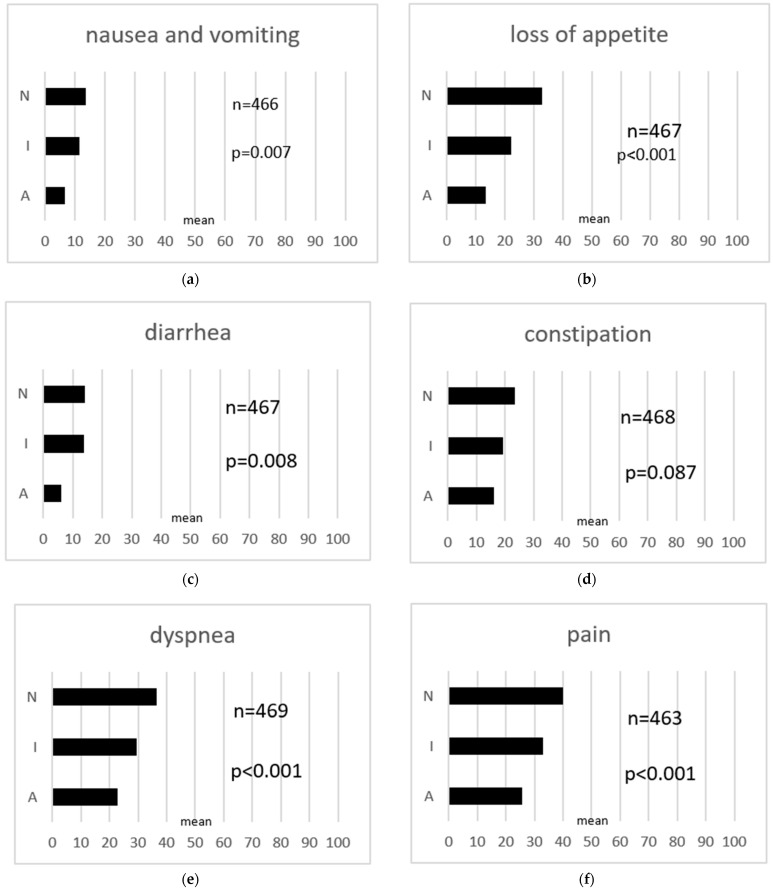
(**a–n**) EORTC QLQ-C30, association with quality of life. N: patients who had no interest in sex and were not sexually active; I: patients who had an interest in sex but were not sexually active; A: patients who had an interest in sex and who were sexually active.

**Figure 2 cancers-16-00811-f002:**
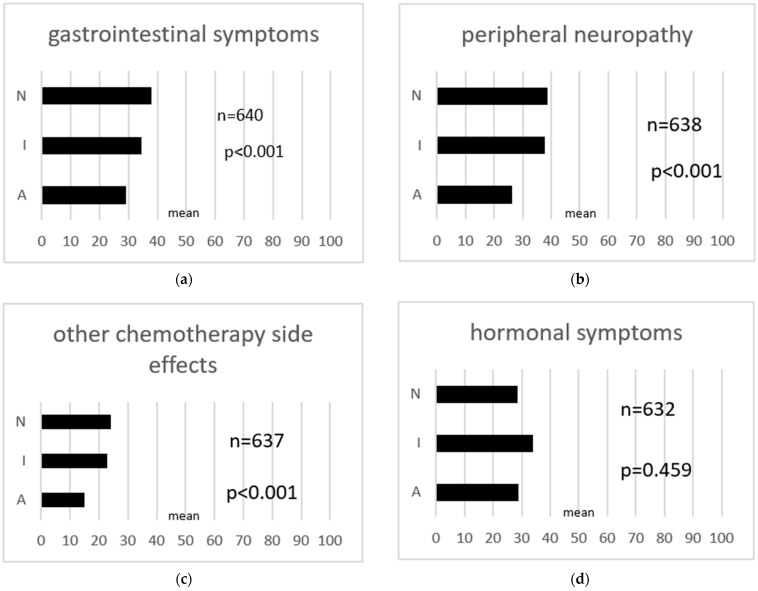
(**a–f**) EORTC QLQ-OV28, association with quality of life. N: patients who had no interest in sex and were not sexually active; I: patients who had an interest in sex but were not sexually active; A: patients who had an interest in sex and who were sexually active.

**Figure 3 cancers-16-00811-f003:**
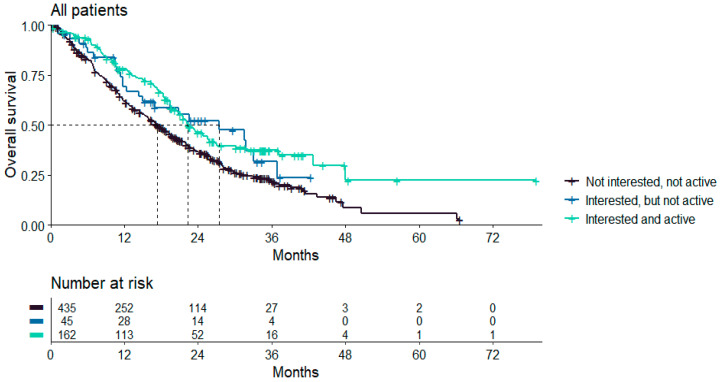
Kaplan–Meier; associations with overall survival.

**Table 1 cancers-16-00811-t001:** Patient characteristics.

	No. of Patients n	Group A ^1^ n	Group I ^2^ n	Group N ^3^ n	Significance
**Median age at first diagnosis in years (range)**	644	162 53 years (24–80)	45 58 years (range: 35–76)	437 62 years (range: 32–85)	*p* < 0.001
**Median age at randomization** **in years (range)**	664	162 57 years (25–81)	45 60 years (37–78)	437 65 years (33–87)	*p* < 0.001
**ECOG 0 ***	298	94 (31.5%)	20 (6.7%)	184 (61.7%)	
**ECOG 1 ***	300	61 (20.3%)	24 (8.0%)	215 (71.7%)	*p* < 0.001
**ECOG 2 ***	43	6 (14.0%)	1 (2.3%)	36 (83.7%)	
**No. of recurrences** **1**	416	115 (27.6%)	35 (8.4%)	266 (63.9%)	
**No. of recurrences** **2**	164	41 (25.0%)	7 (4.3%)	116 (70.7%)	*p* < 0.001
**No. of recurrences** **>2**	64	6 (9.4%)	3 (4.7%)	55 (85.9%)	
**Presence of Ascites**	168	30 (17.9%)	12 (7.1%)	126 (75%)	*p* = 0.002
**BMI** **(median)**	644	162 (24.8)	45 (25.2)	437 (25.8)	*p* = 0.282
**FIGO** **I + II**	71	20 (28.2%)	8 (11.3%)	43 (60.6%)	
**FIGO** **III**	444	106 (23.9%)	31 (7.0%)	307 (69.1%)	*p* = 0.438
**FIGO** **IV**	108	29 (26.9%)	4 (3.7%)	75 (69.4%)	
**Grading** **G1**	27	12 (44.4%)	0 (0.0%)	15 (55.6%)	
**Grading** **G2**	183	47 (25.7%)	15 (8.2%)	121 (66.1%)	*p* = 0.182
**Grading** **G3**	384	91 (23.7%)	27 (7.0%)	266 (69.3%)	

Patient characteristics. ^1^: Interest in sex and active. ^2^: Interest in sex and not active. ^3^: No interest in sex and not active. * At randomization.

## Data Availability

For scientists, access to our data is possible. Contact NOGGO.de.

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
