# Peer review of "Sexuality as a Prognostic Factor—Results of an Individual Patient Data NOGGO (North-Eastern German Society of Gynecological Oncology)-Meta-Analysis of 644 Recurrent Ovarian Cancer Patients Prior to Chemotherapy"

_cancers, 2024, doi:10.3390/cancers16040811_

Round 1
Reviewer 1 Report (Previous Reviewer 3)
Comments and Suggestions for Authors
Dear Authors
I have no additional comments or enquires regarding the manuscript.
This manuscript is a resubmission of an earlier submission. The following is a list of the peer review reports and author responses from that submission.
Round 1
Reviewer 1 Report
Comments and Suggestions for Authors
Paper about sexual interest/activity and survival of patients with recurrent OC.
The selection bias should be clearly cited. Some trials included elderly patients, others platinum resistant recurrence. The mean age at inclusion is probably different and age is an important factor for sexual interest/activity...
The relation between sexual interest/activity and survival should be better discussed. Some could think that sexual activity increases survival. The realty is that patients with sexual interest/activity have global better general status than others and logically survive longer.
these two points should be clearly added in the discussion.
However, we share the conclusion that sexual health should be intergrated in the overall management of cancer patients.
Reviewer 2 Report
Comments and Suggestions for Authors
Paper including more patients data already published by the Authors: 10.1200/JCO.2023.41.16_suppl.e24128
Reviewer 3 Report
Comments and Suggestions for Authors
Dear Authors
Thank you for very interesting manuscript in the field which is still not covered properly especially in oncology, supportive and palliative care.
As I liked the manuscript upon reading still I have one huge concern- most of the datas show that patients from group A (somehow the youngest and with best ECOG in comparison to other groups) have highest overall QoL with better physical, social and emotional functionality and the lowest rates of many important symptoms associated with the ovarian cancer and its treatment (nausea and vomiting, anorexia, diarrhea, dyspnea, pain and fatigue) and afterall have achieved the longest overall survival. In discussion and conclusions Authors are convincing that all of above is caused by interest in sex and sexual activity and that sexuality is a marker for quality of life and for overall survival.
But what I am fearing of is that it is exactly the opposite- patients were interested in sex and active because they had better performance status and much less severe symptoms (like neuropathic pain, all the gastrointestinal etc). This group of patients was just younger and in much much better condition than the others (longer survival and lower attitiude towards disease and treatment sic!).
And one small thing- I do not understand what is a rule of presenting the references in the manuscript: nither due to alphabetic order, nor due to citation order.